# Efficiency Optimization in Multi-Branch Converters through Dynamic Control

**Marek Pavlík** [1], **Matej Bereš** [2,*], **Dobroslav Kováč** [2], **Tibor Vince** [2], **Irena Kováčová** [2] **and Ján Molnár** [2]

[1]  Department of Electric Power Engineering, Technical University of Košice, Letná 1/9, Košice-Sever,
    042 00 Košice, Slovakia; marek.pavlik@tuke.sk
[2]  Department of Theoretical and Industrial Electrical Engineering, Technical University of Košice, Letná 1/9,
    Košice-Sever, 042 00 Košice, Slovakia; tibor.vince@tuke.sk (T.V.)
*   Correspondence: matej.beres@tuke.sk

**Abstract:** As the global emphasis on solar energy intensifies, optimizing the efficiency of photovoltaic panels becomes crucial in meeting energy demands sustainably. Addressing this, our research delves deeply into advancing maximum power point tracking (MPPT), a pivotal component in perfecting the energy conversion process. Leveraging state-of-the-art mathematical modeling, in-depth simulations, and comprehensive experimental validation, we set out to markedly refine the performance of non-isolated multi-branch buck DC–DC converters. In this pursuit, we introduce an innovative algorithm meticulously designed to adjust the number of active branches. This adjustment is rooted in robust efficiency metrics, ensuring optimal power delivery even under dynamic and fluctuating conditions. We place a distinct emphasis on the transformative role of current in determining converter efficiency. Drawing from our findings, we advocate for an adaptive control strategy, precisely engineered to thrive in a spectrum of operational contexts. With this study, we not only present pivotal contributions to the domain of photovoltaic technology but also chart out clear expectations for future endeavors. Our hope is that these advancements serve as foundational steps, guiding the evolution of sustainable energy generation.

**Keywords:** photovoltaic panel; converter for photovoltaic; multi-branch converter; MPP tracking; sustainability; efficiency enhancement





## 1. Introduction

As the global demand for sustainable energy generation continues to grow, harnessing the power of photovoltaic panels has become increasingly significant. The deployment of photovoltaic power plants worldwide highlights the urgent need to optimize the efficiency of these panels, which convert sunlight into electrical power. While previous research has delved into maximum power point tracking (MPPT) methodologies, our study sets out to pioneer innovative approaches that adapt to dynamic operational conditions, a novel contribution that we will explicitly emphasize in this introduction.

Throughout this work, we will uncover the inherent losses within multi-branch buck DC–DC converters, shedding light on the suboptimal activation of all branches simultaneously. This revelation underscores the critical need for dynamic control strategies, a novel approach, which we will explicitly highlight, aimed at adjusting the number of active branches to suit diverse operational conditions. Moreover, our findings place a novel emphasis on the importance of precise current control over power value, thus reshaping the converter's efficiency landscape, another key novelty that we will discuss.

Our innovative algorithm, as evidenced by visual representations, demonstrates its ability to dynamically optimize converter configurations, a novel approach we will underscore, achieving enhanced efficiency in varying input power conditions and diverse scenarios. This study bridges the gap between branch number and system performance,

emphasizing adaptability, a unique contribution that we will clearly delineate, for optimal power delivery and efficiency in differing operational circumstances.

In this introduction, we aim to explicitly highlight these novel contributions, paving the way for a comprehensive understanding of how our research contributes to advancing photovoltaic technology and sustainable energy generation.

The number of photovoltaic power plants in the world is increasing every year. It can be assumed that this trend will continue. The installation of renewable energy sources, including photovoltaic power plants, is expected to contribute to reducing the carbon footprint. While the efficiency of photovoltaic panels continues to improve, it still does not reach high values. Currently, the efficiency of photovoltaic panels is defined at values between 18–24 percent. Therefore, photovoltaic panels utilize approximately a quarter of the solar radiation. Energy from photovoltaic panels is converted into AC power in the converter, also with a certain efficiency. Every electrical device operates at a certain efficiency, so it is important to increase the efficiency of each device whenever possible. The converter has the ability to track the maximum power point of the photovoltaic panel. The power of the photovoltaic panel is determined as the product of current and voltage. Each panel has its own operating point at which its power output is the highest. This point is marked as the maximum power point (MPP). Several different methods are used for tracking the MPP. The first option is without tracking the MPP point. This is the most basic and simplest solution. The output voltage is set to the value $V_{ref}$, which is determined from previous measurements for a given photovoltaic source. This solution assumes that $V_{ref}$ is equal to $V_{MPP}$. However, this variant does not consider the temperature change and solar radiation changes on the position of the MPP, and the output voltage is still set to $V_{ref}$. Therefore, this solution achieves the worst results. In the past, the output voltage $V_{ref}$ was adjusted according to the season, but this approach is associated with many problems [1–6].

One part of the methods consists of so-called indirect methods, which are characterized by the use of average values and estimates. Also, this variant cannot be considered accurate and does not achieve as good results as more advanced methods. In the open-circuit voltage method, it is utilized that $V_{MPP} = k \times V_{OC}$.

The constant $k$ typically ranges from 0.73 to 0.8 for silicon photovoltaic panels. However, this method never directly finds the MPP point, and therefore, it is not very precise. On the other hand, the range of values for the coefficient $k$ is not large, so it remains close to the MPP point. Based on the requirement $V_{MPP} = k \times V_{OC}$, it is evident that $V_{OC}$ must always be measured. However, this is only possible when the circuit is disconnected. For this reason, a test cell is placed on the photovoltaic panel, which is used solely for measuring the required values. The principle of another method (short-circuit current) is similar to the open-circuit voltage method. The only difference is that the short-circuit current $I_{SC}$ is measured instead of the open-circuit voltage $V_{OC}$. The formula for $I_{MPP}$ is similar, and it is $I_{MPP} = k \times I_{SC}$. In this method, the coefficient $k$ is defined in the range of 0.78 to 0.9. The temperature gradient method utilizes the fact that the open-circuit voltage $V_{OC}$ is linearly dependent on the temperature of the cell. By using (1) and measuring the temperature, it is possible to calculate $V_{OC}$.

$$V_{OC} = V_{OC(STC)} + (T - T_{STC}) \cdot \frac{dV_{OC}}{dT} \tag{1}$$

where $V_{OC}$ is open-circuit voltage, $V_{OC(STC)}$ is open-circuit voltage for standard temperature conditions, $T$ is the temperature of the cell, $T_{STC}$ is the temperature of the cell for standard temperature conditions, $dV_{OC}/dT$ is thermal gradient.

$V_{MPP}$ is then calculated similarly to previous methods, $V_{MPP} = k \times V_{OC}$. The measured variables are $V$ and $T$. The temperature parametric method is similar, but it calculates $V_{MPP}$ using (2) [6]:

$$V_{MPP} = (u + G \cdot v) - T \cdot (w + G \cdot y) \tag{2}$$

where $V_{MPP}$ is voltage for maximum power point, $T$ is the temperature of the cell, $G$ is solar radiation intensity, and $u$, $v$, $w$, $y$ are PV panel parameters for different G.

The measured variables in this method are $u$, $G$, and $T$.

Direct methods utilize the fact that the power–voltage characteristic has only one maximum, and thus, the derivative is equal to 0. They also take advantage of the fact that to the left of the maximum, the curve of the function is increasing, and to the right of the maximum, the function is decreasing. These methods are based on measuring $V$ and $I$. Algorithms that are more sophisticated are used for tracking the MPP in these methods compared to indirect methods. These methods achieve higher accuracy in finding the MPP point and, therefore, maximize the delivered electrical energy. However, the disadvantage is that they are more complex to implement compared to indirect methods and, consequently, have a higher cost [7–10].

*Perturb and Observe—Classic Method*

The perturb and observe (P&O) method is a classic approach that relies on the shape of the power–voltage curve. There is only one maximum point. Values of voltages lower than $V_{MPP}$ result in an increasing curve, and values higher than $V_{MPP}$ result in a decreasing curve. So, if increasing the voltage leads to an increase in power, then the algorithm assumes it is moving to the left of the MPP point in the next iteration. By successive iterations, the algorithm reaches a point where the assumption that increasing voltage increases power no longer holds. At this point, the algorithm is to the right of the MPP point. Then, the algorithm stops increasing the voltage and, conversely, starts decreasing it. If decreasing the voltage leads to an increase in power, the algorithm continues to reduce the voltage until further voltage reduction results in decreased power. This process repeats, and the algorithm converges to the MPP point in this way.

One negative of this method is that during rapid changes in radiation intensity, convergence can be slow. Another negative is that the algorithm never finds the MPP point directly but always moves around it. The algorithm creates a regular shift in voltage by a certain value $\Delta V$ and evaluates the power change. Based on this data, the algorithm decides the direction of the next voltage shift. This process repeats continuously. A problem can arise with low solar radiation intensity values when the power–voltage curve is very flat, making it challenging for the algorithm to find its maximum. Nevertheless, it is one of the most commonly used methods [8,9,11].

Another variation of the P&O method is the so-called three-point method, which uses three points for tracking the MPP instead of two. The first point is considered the current state of tracking. Then, there is a shift to the working point (the second point), followed by a shift to point C (the third point). The shift to point C is twice as much as in the previous shift but in the opposite direction [9,12].

Advanced methods require higher computational power. For example, the fuzzy logic control (FLC) method uses fuzzy logic. In the process of finding the maximum power point (MPP), fuzzification is used, which involves transforming input numerical data (measured current and voltage values) into linguistic variables. The outputs are two variables: error ($E$) and error change ($\Delta E$).

The artificial neural network (ANN) method uses an artificial intelligence algorithm consisting of three layers: input, hidden, and output. The input layer includes measured input variables such as voltage, current, temperature, and solar radiation intensity. In the second layer, calculations are performed based on weighted connections between nodes. This type of algorithm is capable of machine learning, and over time, it adjusts the weights of connections between various nodes based on measured data. This results in better tracking and estimation of the MPP occurrence.

However, a disadvantage of this method is that it needs to be configured and is applicable only to specific locations and panel types. Another issue arises when the characteristics of the photovoltaic panel change. Calibration of the system is required after a certain period [11–14].

The problems with finding the MPP (maximum power point) are so significant that there are numerous publications dedicated to them. Some of these describe research in

the transition of MPPT (maximum power point tracking) search to the flexible power point tracking (FPPT) method. Under normal operating conditions, photovoltaic systems always operate on MPfiPT. However, in the event of a grid disturbance or changes in light intensity, it is necessary to operate with the FPPT system. Changes in solar radiation have a significant impact on MPPT, and therefore, in such situations, it is necessary to shift the operating point to FPPT on the left or right side of the power–voltage curve. In the publication [15], the authors propose a single fuzzy logic-based algorithm. The authors demonstrate that their algorithm for the controller forces the photovoltaic panel to operate very close to its optimal trajectory of maximum power, even in the presence of changes in solar radiation. The authors in their publications present similar results [15–18].

In paper [19], the authors propose an algorithm based on the variable-step conductance increment method and pseudo-MPP correction. This algorithm also enhances the MPP search in response to changes in solar radiation. From the studied publications, there is a strong emphasis on achieving the most precise MPP tracking over time. Any improvement in the tracking represents an increase in the efficiency of the converter's operation and enhances the conversion of DC electrical energy to AC [17–19].

The article is structured into multiple sections, each addressing distinct aspects of the study. The first section delves into the mathematical model, where fundamental principles, parameters, and parasitic values of various converter components are employed to derive an equation [19]. This equation determines the optimal number of branches in the converter for a given input voltage and current. The second section focuses on the validation of these calculations through a simulation program, elucidating the procedure and simulation methodology. The third section is dedicated to the practical validation of the mathematical and simulation models through measurements. The fourth section revolves around the design and validation of a control algorithm, which can serve as an extension to existing methodologies. This structured approach allows for a comprehensive exploration of the converter's behavior and efficiency optimization across different facets of the study [19–22].

## 2. Mathematical Analysis of a Non-Isolated Multi-Branch Buck DC–DC Converter and Calculated Results

The schematic diagram illustrating the configuration of the multi-branch converter with $n$ branches is presented in Figure 1. Figure 1 displays both the fundamental circuit (Figure 1a) and the circuit consisting of parasitic electrical components of the converter (Figure 1b). In this analysis, we exclusively focus on the operation of the converter in the buck mode since a cascaded DC–DC buck–boost converter topology is used, with the omission of the second transistor. Nevertheless, for the purpose of efficiency assessment, it is necessary to include diodes $D_{12}$ to $D_{n2}$. It is important to note that the analysis does not encompass an examination of the boost mode, which is beyond the scope of this manuscript. As elucidated later, the calculations can be readily adapted to attain the final mathematical representation for any topology required for a specific application. It should be emphasized that both simulations and experimental results encompass the inclusion of the aforementioned diodes, and thus, the analysis takes these diodes into account. The diode's presence does not invalidate the subsequent analysis proposed herein, nor does it impact the validity of the simulation or experimental findings [19,21–27].

Each electrical component that is denoted by the index $P$ corresponds to the parasitic element within a practical model of the converter. In the context of electrical current, the first Kirchhoff's law, denoted as (3), is employed [21–25,28].

$$i = \sum_{j=1}^{n} i_j \tag{3}$$

Each individual branch of the converter operates at identical switching frequencies and duty cycles, synchronized with those of the other branches. The control signals are temporally staggered, determined by $T/n$, where $T$ represents the switching period and

*n* denotes the number of converter branches. Using these phase-shifted control signals ensures consistent interleaving whenever the number of branches changes [25,28–30].

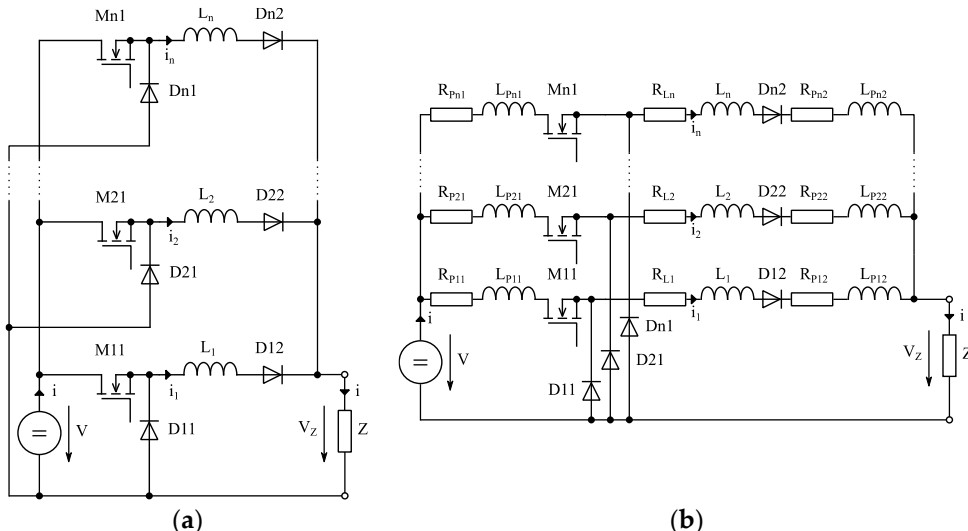

**Figure 1.** The schematic representation illustrates the interconnection of an *n*-branched DC–DC buck converter, where (**a**) represents the fundamental circuit and (**b**) depicts the circuit consisting of parasitic electrical components.

### 2.1. Analysis of Energy Accumulation in the Initial Time Period

In this operational phase, the switching device, specifically a MOSFET transistor, is closed. This results in energy buildup in the circuit's primary inductive element, as shown in Figure 2.

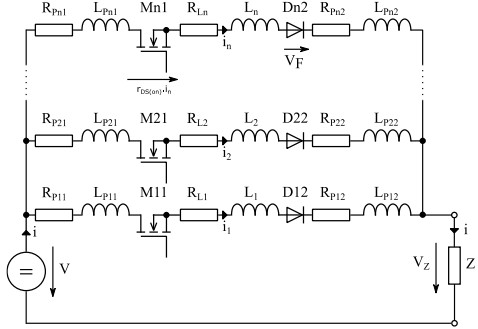

**Figure 2.** The replaced circuit configuration during the initial operational phase of the converter.

The second Kirchhoff's law, as denoted by (4), can be applied to each of the *n* conductive branches of the converter.

$$
\begin{aligned}
-V + r_{DS(on)n} \cdot i_n + R_{Pn1} \cdot i_n + L_{Pn1} \cdot \frac{di_n}{dt} \\
+ V_F + L_n \cdot \frac{di_n}{dt} + R_{Ln} \cdot i_n + R_{Pn2} \cdot i_n \\
+ V_Z + L_{Pn2} \cdot \frac{di_n}{dt} = 0
\end{aligned}
\tag{4}
$$

here, $V$ stands for the input voltage, while $R_{Pn1}$ and $L_{Pn1}$ denote the parasitic resistance and inductance of the input conductor, respectively. $r_{DS(on)}$ denotes the resistance of the MOSFET transistor when it is in the "on" state, while $R_{Ln}$ and $L_n$ represent the resistance and inductance of the primary inductor. $V_F$ signifies the voltage drop across a diode, $R_{Pn2}$ is the resistance of the output conductor connected to the load, and $V_Z$ represents the load voltage [19,23–25,28–31].

Due to the resistive and inductive loads, the current waveform in continuous conduction mode (CCM) takes on a distinct shape, as shown in Figure 3.

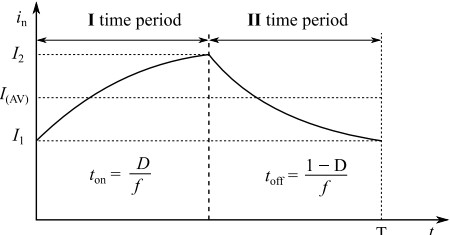

**Figure 3.** Current waveform in the continuous conduction mode (CCM) of the converter with $n$ branches. The current waveform, displayed in one branch, shows that $I_1$ is the minimum current in continuous conduction mode, while $I_2$ is the maximum current in this mode. The same current waveform applies to $n$ branches.

In accordance with Figure 2, the loop circuit can be simplified through parameter consolidation as per (5) to (8) [19,32].

$$V_Z = Z \cdot I = Z \cdot \sum_{i=1}^{n} i_i \approx Z \cdot n \cdot i_n \tag{5}$$

$$V_F = r_F \cdot i_n + V_{TO} \tag{6}$$

$$R_{n1} = R_{Pn1} + R_{Pn2} + r_F + r_{DS(on)} + R_{Ln} \tag{7}$$

$$L_{n1} = L_{Pn1} + L_n + L_{Pn2} \tag{8}$$

here, $V_{TO}$ is the diode's threshold voltage, $R_{n1}$ is the resistance of the $n$-th branch in the first time period, $r_F$ stands for the diode's forward resistance, and $L_{n1}$ indicates the inductance of the $n$-th branch converter during that initial period. Utilizing (5) to (8), Equation (4) can be adjusted as presented in (9) [22–25,28]:

$$V_{TO} - V + Z \cdot n \cdot i_n + R_{n1} \cdot i_n + L_{n1} \cdot \frac{di_n}{dt} = 0 \tag{9}$$

When (9) is restructured using a mathematical shape operator, the equation takes the following form [19,32]:

$$V_{TO} - V + Z \cdot n \cdot \hat{i}_n + R_{n1} \cdot \hat{i}_n + pL_{n1} \cdot \hat{i}_n - pL_{n1} \cdot I_1 = 0 \tag{10}$$

Beginning with (10), the current can be depicted in the manner described in [19]:

$$\hat{i}_n = \frac{V - V_{TO}}{(R_{n1} + n \cdot Z)} \cdot \frac{\frac{(R_{n1} + n \cdot Z)}{L_{n1}}}{\frac{(R_{n1} + n \cdot Z)}{L_{n1}} + p} + I_1 \cdot \frac{p}{\frac{(R_{n1} + n \cdot Z)}{L_{n1}} + p} \tag{11}$$

By performing an inverse transformation of (11) back into the time domain, we derive (12) [21–23].

$$i_n = \frac{V - V_{TO}}{(R_{n1} + n \cdot Z)} \cdot \left(1 - e^{-\frac{(R_{n1} + n \cdot Z)}{L_{n1}} \cdot t}\right) + I_1 \cdot e^{-\frac{(R_{n1} + n \cdot Z)}{L_{n1}} \cdot t} \tag{12}$$

When we replace $t$ with $t_{on} = D/f$ in the time domain expression for the current, at a specific time instance, it assumes the value of $I_2$, as illustrated in Figure 2. The resultant equation for the current $I_2$ can be expressed as follows (13) [21–23]:

$$I_2 = \frac{V - V_{TO}}{(R_{n1} + n \cdot Z)} \cdot \left(1 - e^{-\frac{(R_{n1} + n \cdot Z)}{L_{n1}} \cdot \frac{D}{f}}\right) + I_1 \cdot e^{-\frac{(R_{n1} + n \cdot Z)}{L_{n1}} \cdot \frac{D}{f}} \tag{13}$$

here, $D$ signifies the duty cycle, and f corresponds to the switching frequency.

Next, to ascertain the value of $I_1$, we will delve into the converter's second time period.

### 2.2. Examination of the Second Time Period: Energy Delivery to the Load

In the converter's second operational phase, the switching device deactivates, allowing the energy stored in the primary inductor to flow to the load through diodes $D_{n1}$ and $D_{n2}$, as shown in Figure 4.

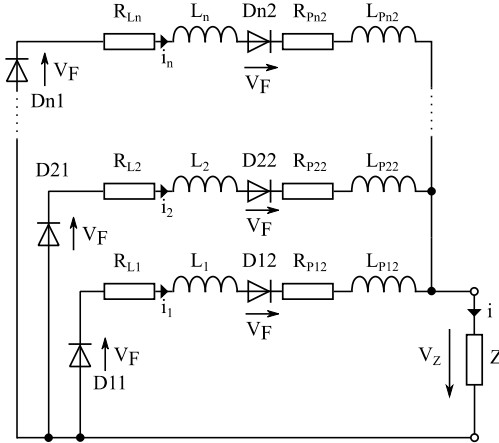

**Figure 4.** Substituted circuit layout for the converter's second operational phase.

For every conductive branch, $n$, in the converter, the second Kirchhoff's law is applicable as articulated in (14) [23–25].

$$V_Z + V_F + L_n \cdot \frac{di_n}{dt} + R_{Ln} \cdot i_n$$
$$+ L_{Pn2} \cdot \frac{di_n}{dt} + V_F + R_{Pn2} \cdot i_n = 0 \tag{14}$$

In addition to (5), the following equations can also be employed to simplify the loop circuit through parameter consolidation [23–25]:

$$R_{n2} = R_{Ln} + 2 \cdot r_F + R_{Pn2} \tag{15}$$

$$L_{n2} = L_n + L_{Pn2} \tag{16}$$

Utilizing equations for parameter consolidation, (14) can be transformed into the following expression [23–25]:

$$2 \cdot V_{TO} + L_{n2} \cdot \frac{di_n}{dt} + R_{n2} \cdot i_n + n \cdot Z \cdot i_n = 0 \tag{17}$$

When (17) is recast using a mathematical shaping operator, the equation will adopt the subsequent format [23–25]:

$$2 \cdot V_{TO} + pL_{n2} \cdot \hat{i}_n + n \cdot Z \cdot \hat{i}_n - pL_{n2} \cdot I_2 + R_{n2} \cdot \hat{i}_n = 0 \tag{18}$$

Now, based on (18), the current can be represented in the following format [23–25]:

$$\hat{i}_n = \frac{-2 \cdot U_{TO}}{(R_{n2} + n \cdot Z)} \cdot \frac{\frac{(R_{n2} + n \cdot Z)}{L_{n2}}}{\frac{(R_{n2} + n \cdot Z)}{L_{n2}} + p} + I_2 \cdot \frac{p}{\frac{(R_{n2} + n \cdot Z)}{L_{n2}} + p} \tag{19}$$

By performing an inverse transformation of (17) back into the time domain, we derive (20) [23–25].

$$i_n = \frac{-2 \cdot V_{TO}}{(R_{n2} + n \cdot Z)} \cdot \left(1 - e^{-\frac{(R_{n2}+n\cdot Z)}{L_{n2}} \cdot t}\right) + I_2 \cdot e^{-\frac{(R_{n2}+n\cdot Z)}{L_{n2}} \cdot t} \tag{20}$$

If we substitute $t$ with $t_{off} = (1 - D)/f$ in the time domain expression of the current, it will provide a current $I_1$ at a specific time instance, as illustrated in Figure 2. The derived equation representing the current $I_1$ is presented below [23–25]:

$$I_1 = \frac{-2 \cdot V_{TO}}{(R_{n2} + n \cdot Z)} \cdot \left(1 - e^{-\frac{(R_{n2}+n\cdot Z)}{L_{n2}} \cdot \frac{(1-D)}{f}}\right) + I_2 \cdot e^{-\frac{(R_{n2}+n\cdot Z)}{L_{n2}} \cdot \frac{(1-D)}{f}} \tag{21}$$

### 2.3. Calculation of the Total Power Loss

Utilizing (13) and (21), it is feasible to derive the current values of $I_1$ and $I_2$ at the commencement of each time period. Next, by substituting the computed value of $I_2$ from (13) into (21), we obtain the following outcome [23–25]:

$$\begin{aligned} I_1 = \frac{-2 \cdot V_{TO}}{(R_{n2}+n\cdot Z)} \cdot \left(1 - e^{-\frac{(R_{n2}+n\cdot Z)}{L_{n2}} \cdot \frac{(1-D)}{f}}\right) \\ + \left(\frac{V - V_{TO}}{R_{n1}+n\cdot Z} \cdot \left(1 - e^{-\frac{(R_{n1}+n\cdot Z)}{L_{n1}} \cdot \frac{D}{f}}\right) + I_1 \cdot e^{-\frac{(R_{n1}+n\cdot Z)}{L_{n1}} \cdot \frac{D}{f}}\right) \cdot e^{-\frac{(R_{n2}+n\cdot Z)}{L_{n2}} \cdot \frac{(1-D)}{f}} \end{aligned} \tag{22}$$

The ultimate expression for the current value $I_1$ is attained by transforming (22) into the following format [23–25]:

$$I_1 = \frac{\frac{-2 \cdot V_{TO}}{(R_{n2}+n\cdot Z)} \cdot \left(1 - e^{-\frac{(R_{n2}+n\cdot Z)}{L_{n2}} \cdot \frac{(1-D)}{f}}\right) + \left(\frac{(V-V_{TO})}{(R_{n1}+n\cdot Z)} \cdot \left(1 - e^{-\frac{(R_{n1}+n\cdot Z)}{L_{n1}} \cdot \frac{D}{f}}\right) \cdot e^{-\frac{(R_{n2}+n\cdot Z)}{L_{n2}} \cdot \frac{(1-D)}{f}}\right)}{\left(1 - e^{-\frac{(R_{n1}+n\cdot Z)}{L_{n1}} \cdot \frac{D}{f}} \cdot e^{-\frac{(R_{n2}+n\cdot Z)}{L_{n2}} \cdot \frac{(1-D)}{f}}\right)} \tag{23}$$

The expression for the current value $I_2$ can be obtained by replacing (23) with (13), leading to the following result [23–25]:

$$\begin{aligned} I_2 = \frac{V - V_{TO}}{(R_{n1}+n\cdot Z)} \cdot \left(1 - e^{-\frac{(R_{n1}+nZ)}{L_{n1}} \cdot \frac{z}{f}}\right) \\ + \left(\frac{\frac{-2 \cdot V_{TO}}{(R_{n2}+n\cdot Z)} \cdot \left(1 - e^{-\frac{(R_{n2}+n\cdot Z)}{L_{n2}} \cdot \frac{(1-D)}{f}}\right) + \left(\frac{V-V_{TO}}{(R_{n1}+n\cdot Z)} \cdot \left(1 - e^{-\frac{(R_{n1}+n\cdot Z)}{L_{n1}} \cdot \frac{D}{f}}\right) \cdot e^{-\frac{(R_{n2}+n\cdot Z)}{L_{n2}} \cdot \frac{(1-D)}{f}}\right)}{\left(1 - e^{-\frac{(R_{n1}+n\cdot Z)}{L_{n1}} \cdot \frac{D}{f}} \cdot e^{-\frac{(R_{n2}+nZ)}{L_{n2}} \cdot \frac{(1-D)}{f}}\right)}\right) \cdot e^{-\frac{(R_{n1}+n\cdot Z)}{L_{n1}} \cdot \frac{D}{f}} \end{aligned} \tag{24}$$

Using Equations (12), (20), (23), and (24), the current waveform for any converter branch can be determined.

To determine the ideal number of branches for the converter, a study was undertaken to evaluate power losses linked with varying branch counts. These losses include conduction losses, static losses in both time intervals, and switching losses in the power converter. The considered losses in a power converter include those produced by semiconductor switches (like MOSFETs and diodes) and passive elements (such as inductors).To pinpoint the optimal number of branches for the converter, an exhaustive power loss analysis was undertaken. These losses encompassed conduction losses such as $P_{RPn1}$ (from current passing through the input wire resistance of the *n*-th converter branch), $P_{MF}$ (from the MOSFET transistor's conductive state), $P_{MR}$ (MOSFET's non-conductive or quiescent losses), $P_{RLn}$ (from current through the inductor wire resistance of the *n*-th branch), $P_{DF}$ (when the diode is conductive), $P_{DR}$ (diode's non-conductive or quiescent losses), and $P_{RPn2}$ (from current through the output wire resistance of the *n*-th branch). Switching losses also factored in, including $P_{Mon}$ (during MOSFET turn-on), $P_{Moff}$ (during MOSFET turn-off),

$P_{Doff}$ (diode turn-off), and $P_{Don}$ (diode turn-on). Leveraging standardized waveforms, semiconductor switching periods, and Equation (3), the cumulative power losses, termed $P_W$, were computed as shown in (25) [19–24].

$$
\begin{aligned}
P_W = P_{DR} + P_{MF} + P_{MR} + P_{DF} + P_{RPn2} + P_{Mon} \\
+ P_{Moff} + P_{Doff} + P_{Don} + P_{RPn1} + P_{RLn}
\end{aligned}
\tag{25}
$$

The average current for the *n*-th branch of the converter can be determined as follows:

$$
I_{n(AV)} = \frac{I_1 + I_2}{2} = \frac{I}{n}
\tag{26}
$$

here, $I$ signifies the average current value of the converter. Using Equation (26), the cumulative power losses of the converter can be calculated in the following manner:

$$
\begin{aligned}
P_W = n \cdot V \cdot I_{DS(off)} \cdot (1-D) + n \cdot R_{Pn1} \cdot \left(I_{n(AV)}\right)^2 \cdot D + n \cdot r_{DS(on)} \cdot \left(I_{n(AV)}\right)^2 \cdot D + n \cdot R_{Ln} \cdot \left(I_{n(AV)}\right)^2 \\
+ n \cdot 2 \cdot \left(V_{TO} + r_F \cdot \left(I_{n(AV)}\right)\right) \cdot I_{n(AV)} \cdot (1-D) + n \cdot \left(V_{TO} + r_F \cdot \left(I_{n(AV)}\right)\right) \cdot I_{n(AV)} \cdot D + n \cdot V \cdot I_R \cdot D \\
+ n \cdot R_{Pn2} \cdot \left(I_{n(AV)}\right)^2 + n \cdot 0.5 \cdot V_{FP} \cdot I_2 \cdot t_{fr} \cdot f + n \cdot 0.5 \cdot V \cdot I_2 \cdot t_{off} \cdot f \\
+ n \cdot 0.5 \cdot Q_{rr} \cdot V \cdot f + n \cdot 0.5 \cdot V \cdot I_1 \cdot t_{on} \cdot f
\end{aligned}
\tag{27}
$$

Here, $r_{DS(on)}$ indicates the resistance of the MOSFET channel (from drain to source) when activated, while $r_{DS(off)}$ refers to its resistance when deactivated. $I_R$ denotes the diode's reverse current flow, ton represents the duration for the MOSFET transistor's activation, and toff is the time it requires to deactivate. $Q_{rr}$ is the diode's commutation charge, and $V_{FP}$ is the voltage drop across the diode during its activation period $t_{fr}$. Incorporating Equation (26) into (27) allows for a streamlined representation of the total power losses as detailed in the subsequent formulation [19–25]:

$$
\begin{aligned}
P_W = \left(\frac{I^2}{n}\right) \cdot \left(2 \cdot r_F \cdot (1-D) + R_{Pn1} \cdot D + R_{Ln} + r_F \cdot D + R_{Pn2} + r_{DS(on)} \cdot D\right) + n \cdot V \cdot I_{DS} \cdot (1-D) \\
+ 2 \cdot V_{TO} \cdot I \cdot (1-D) + + n \cdot V \cdot I_R \cdot D + n \cdot 0.5 \cdot V \cdot I_1 \cdot t_{on} \cdot f + n \cdot 0.5 \cdot V \cdot I_2 \cdot t_{off} \cdot f \\
+ n \cdot 0.5 \cdot Q_{rr} \cdot V \cdot f + V_{TO} \cdot I \cdot D + n \cdot 0.5 \cdot V_{FP} \cdot I_2 \cdot t_{fr} \cdot f
\end{aligned}
\tag{28}
$$

The converter's efficiency can be determined as described below [19]:

$$
\eta = \frac{P_{output}}{P_{input}} = \frac{P_{input} - P_W}{P_{input}} = 1 - \frac{P_W}{P_{input}} = 1 - \frac{P_W}{U \cdot I}
\tag{29}
$$

Based on (29), efficiency is largely driven by total power loss, which in turn heavily depends on the specific components of the structure. The overall power losses can also vary with the number of parallel-connected converter branches. When these branches are in parallel, the input current is divided linearly among them, but losses due to each branch's parasitic resistance decrease in a quadratic manner. This suggests that total losses might be reduced with multiple branches compared to just one. However, it is not always accurate to say that adding more branches consistently reduces losses, given the potential for added losses in parallel configurations. This insight paves the way to pinpoint the ideal number of branches for certain input power levels (keeping input voltage and chosen duty cycle constant). Such an understanding permits tweaks in the converter's design to optimize efficiency across different power inputs while retaining core design components and their characteristics [30–32].

Equation (28) needs to be differentiated with respect to *n* in order to identify the optimal number of branches. Differentiating with respect to *n* allows us to find the location

of the local extreme, providing us with the expression for the number of branches that minimizes power loss [30–32].

$$
\begin{aligned}
P'_W = &-\left(\frac{I^2}{n^2}\right) \cdot \left(2 \cdot r_F \cdot (1-D) + R_{Pn1} \cdot D + R_{Ln} + r_F \cdot D + R_{Pn2} + r_{DS(on)} \cdot D\right) \\
&+0.5 \cdot V \cdot I_2 \cdot t_{off} \cdot f + 0.5 \cdot Q_{rr} \cdot V \cdot f + V \cdot I_R \cdot D + 0.5 \cdot V \cdot I_1 \cdot t_{on} \cdot f \\
&+0.5 \cdot V_{FP} \cdot I_2 \cdot t_{fr} \cdot f + V \cdot I_{DS} \cdot (1-D)
\end{aligned}
\tag{30}
$$

When the derivative yields a value of zero, and *n* is expressed, (30) adopts the following format [30–32]:

$$
n = I \cdot \sqrt{\frac{\left(2 \cdot r_F \cdot (1-D) + R_{pn1} \cdot D + R_{Ln} + r_F \cdot D + R_{Pn2} + r_{DS(on)} \cdot D\right)}{0.5 \cdot V \cdot I_2 \cdot t_{off} \cdot f + 0.5 \cdot Q_{rr} \cdot V \cdot f + V \cdot I_{DS} \cdot (1-D) + V \cdot I_R \cdot D + 0.5 \cdot V \cdot I_1 \cdot t_{on} \cdot f + 0.5 \cdot V_{FP} \cdot I_2 \cdot t_{FR} \cdot f}}
\tag{31}
$$

As a result of applying (31), it is possible to pinpoint the best number of branches, taking into account the converter's design and functioning parameters in the buck mode. This fine-tuning ensures the converter effectively transfers maximal power to the load under all operational scenarios. To back this claim, simulation and experimental findings will be showcased in the following portions of this paper [30–32].

### 2.4. Achieved Results Using the Expressed Formulas

The resulting efficiency profiles, illustrated in Figure 5, provide valuable insights into the converter's performance under varying conditions. These profiles were generated through meticulous calculations using MS Excel, which involved applying the expressed mathematical equations. To achieve a comprehensive view of efficiency trends, a macro was strategically employed. This macro systematically adjusted the duty cycle *D*, with precise increments of 0.02, meticulously exploring a broad spectrum of operating scenarios. For each duty cycle setting, the efficiency was computed, considering all possible combinations of individual converter branches, spanning from a single branch up to four branches in total. To conduct these simulations, we relied on the parameter values documented in Table 1, which capture the essential characteristics of the electrical components employed in the converter's design [30–32].

**Calculated resutls of efficiency based on input power**

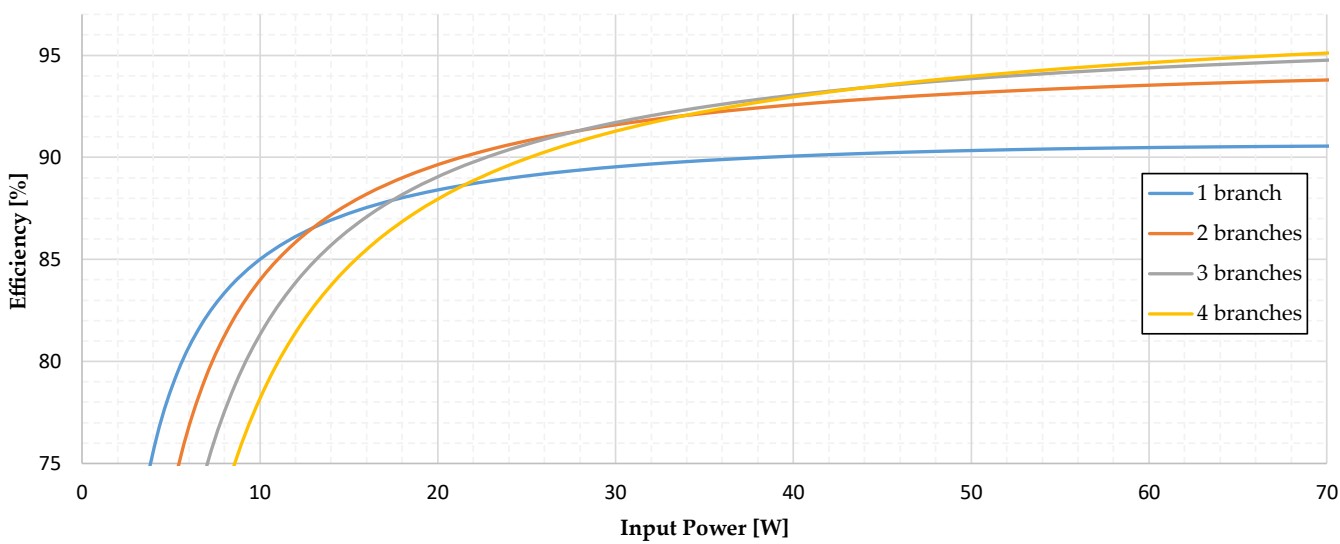

**Figure 5.** The efficiency curves attained through the proposed equations across a range of converter configurations, encompassing varying numbers of branches and input power levels.

**Table 1.** Parameter values for electrical components.

| Parameter | Description | Value |
|---|---|---|
| $n$ | Number of branches | 1–4 |
| $R_{Pn1}$ | Input wire resistance | 0.2 ($\Omega$) |
| $R_{Pn2}$ | Output wire resistance | 0.2 ($\Omega$) |
| $V$ | Input voltage | 30 (V) |
| $R_{DS(on)}$ | Drain–source electrical resistance of the MOSFET when it is in the "on" state | 0.07 ($\Omega$) |
| $R_{Ln}$ | Electrical resistance of the primary inductor | 0.044 ($\Omega$) |
| $r_F$ | Diode resistance in forward direction | 0.02 ($\Omega$) |
| $I_{DS(off)}$ | Leakage current of MOSFET | 250 ($\mu$A) |
| $V_{TO}$ | Diodes' threshold voltage | 0.77 (V) |
| $I_R$ | Reverse current flow in the diode | 2 (mA) |
| $t_{on}$ | Time which transistor takes to turn on | 55 (ns) |
| $t_{off}$ | Time which transistor takes to turn off | 96 (ns) |
| $Q_{rr}$ | Commutation charge of diode | 120 (nC) |
| $V_{FP}$ | The voltage decline across the diode during its turn-on time, $t_{fr}$ | 2 (V) |
| $f$ | frequency | 200 (kHz) |
| $t_{fr}$ | Duration of the voltage decrease on the diode required for its activation | 2 (ns) |
| $D$ | Duty cycle | 0–1 |
| $Z$ | Resistive load | 4.7 ($\Omega$) |

The utilized MOSFET transistor was of the IRF540N variety.

## 3. Simulated Efficiency Results Derived from Variations in the Number of Branches and Input Power Levels

In order to substantiate the accuracy of our calculations, a simulation model was meticulously devised. This simulation model, as depicted in Figure 6, was created using the Proteus simulation environment as our tool of choice. It is worth noting that we employed version 8.9 SP2 of Proteus for this purpose. The Proteus simulation environment relies on ProSpice as its underlying simulation core, ensuring robust and precise simulation capabilities.

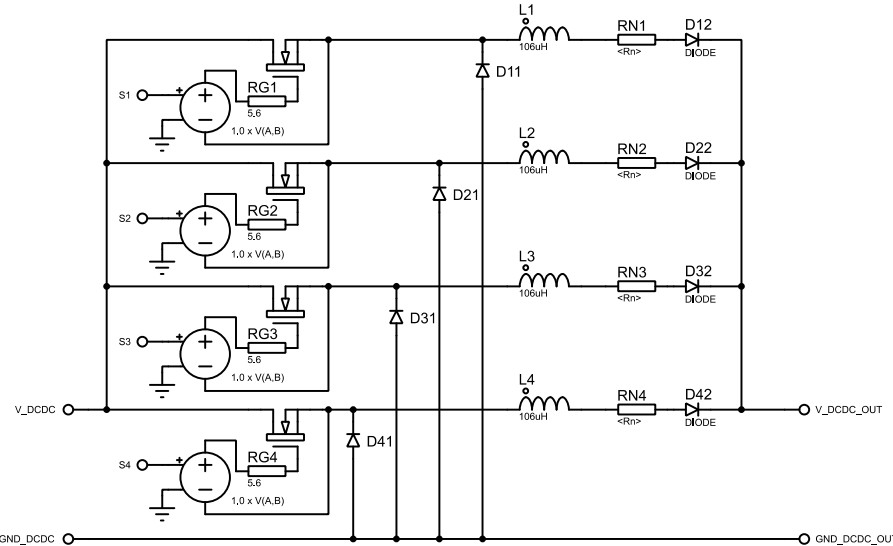

**Figure 6.** The simulation electrical circuit for the four-branch buck converter, accounting for the presence of parasitic resistance.

The utilization of this simulation environment offers a significant advantage, primarily rooted in its ability to seamlessly integrate digital components, like microcontrollers, with analog devices. This versatility becomes particularly advantageous when microcontrollers are programmed within their dedicated programming environments, where their functionalities and algorithms can be fine-tuned and optimized. Subsequently, the compiled

binary files, resulting from the microcontroller programming process, can be effortlessly imported, and employed directly within the simulation environment. This integrated approach ensures a comprehensive and cohesive assessment of digital–analog interactions and system behavior, enhancing the accuracy and depth of our simulations.

In the simulation software, the calculation of input and output power is performed within a carefully designed framework. This framework is composed of essential elements, including current-controlled voltage sources and voltage-controlled voltage sources, combined with an ideal four-quadrant multiplier. These components collectively facilitate the precise calculation of power interactions within the system under study. The outcomes generated by the multipliers are subsequently subjected to further refinement. This is realized using a first-order Laplace low-pass filter transfer function, as depicted in Figure 7. To ensure appropriate filtering and smoothing of the results, a specific value of τ (tau) is configured, which in this context is set to 4 μs. This strategic incorporation of filtering mechanisms not only refines the precision of the simulated results but also aids in removing high-frequency noise or disturbances, ensuring a more accurate representation of the system's behavior.

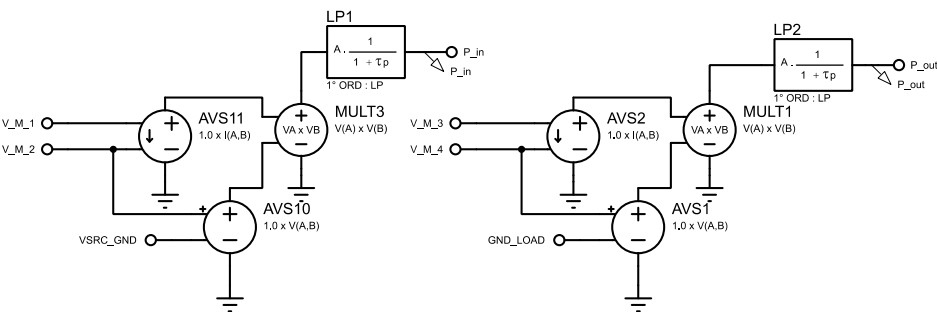

**Figure 7.** Determining input and output power entails the use of ideal current-controlled voltage sources, voltage-controlled voltage sources, voltage multipliers, and first-order Laplace low-pass filter transfer functions.

Following the determination of input and output power, the efficiency can be readily computed, as illustrated in Figure 8. It is important to note that the calculated efficiency result follows the same methodology as that used for input and output power, with the additional step of being filtered through a first-order Laplace low-pass filter transfer function.

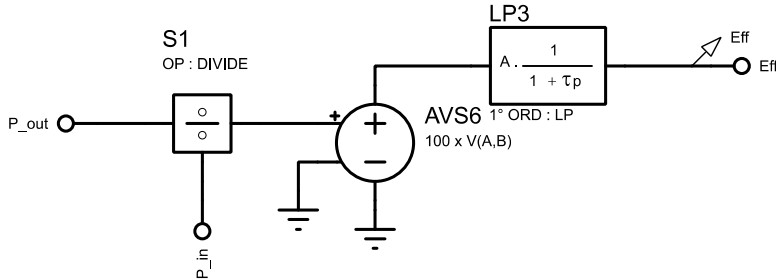

**Figure 8.** The determination of efficiency involves the application of a divider function in conjunction with an arbitrary voltage-controlled voltage source, followed by a refinement process through the use of a first-order Laplace low-pass filter transfer function.

In our simulation, we used the same electrical components as in the preliminary computation phase. Yet, after finalizing the simulation model, significant differences emerged between the computed and simulated outcomes, most prominently in the efficiency figures. This discrepancy may arise from variations in parameter values, wherein the calculation relied on data extracted from manufacturer datasheets rather than real measured values.

Interestingly, when analyzing the results, it becomes apparent that even within the simulation model, optimal efficiency is achieved by varying the number of legs, as visually represented in Figure 9. This finding underscores the significance of adapting the converter's configuration to the specific input power levels to maximize its efficiency.

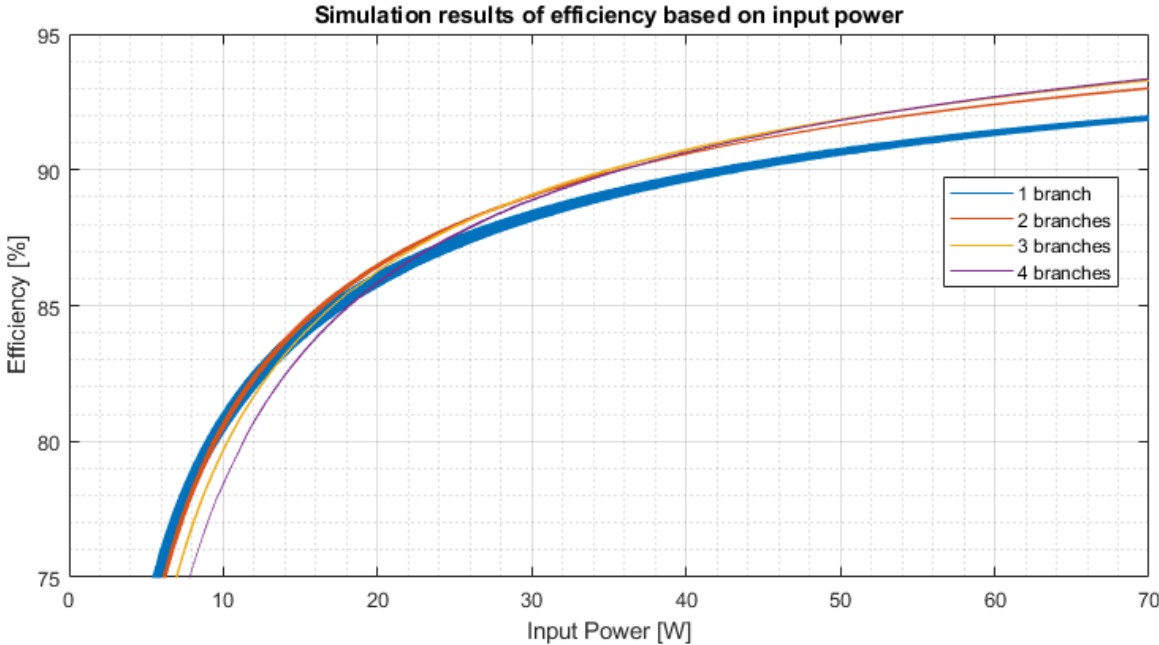

**Figure 9.** Simulation curves obtained through the utilization of a simulation model for the converter.

Leveraging the capabilities of the Proteus simulation environment, we can readily extract and determine the efficiency from the obtained results, thereby enhancing our understanding of the converter's performance characteristics. The results from the simulation program were simulated for each unique combination of converter branch numbers. Subsequently, these results underwent further processing within the Matlab software environment in version of R2023a Update 5 (9.14.0.2337262). This additional step allowed for a streamlined comparison of the individual efficiency outcomes, facilitating a comprehensive assessment of the various configurations.

Nevertheless, the simulation results also affirm that a higher number of branches is not always the optimal choice. Therefore, it is imperative to consider adjusting the converter's number of branches when aiming to supply the load with maximum power while achieving the highest possible efficiency.

## 4. Experimental Efficiency Results Derived from Variations in the Number of Branches and Input Power Levels

In order to validate both the mathematical and simulation results, we proceeded to develop an experimental model. The connection diagram representation of this experimental setup is visually depicted in Figure 10.

Creating an experimental model is a crucial step in the verification process, as it enables us to bridge the gap between theoretical calculations, computer simulations, and real-world performance. The experimental setup provides a tangible means of assessing the practical applicability and accuracy of our derived results, offering valuable insights into the converter's behavior under physical conditions. A tangible implementation of the interleaved converter prototype is depicted in Figure 11. For experimental purposes, it is worth noting that each branch can be physically isolated from one another.

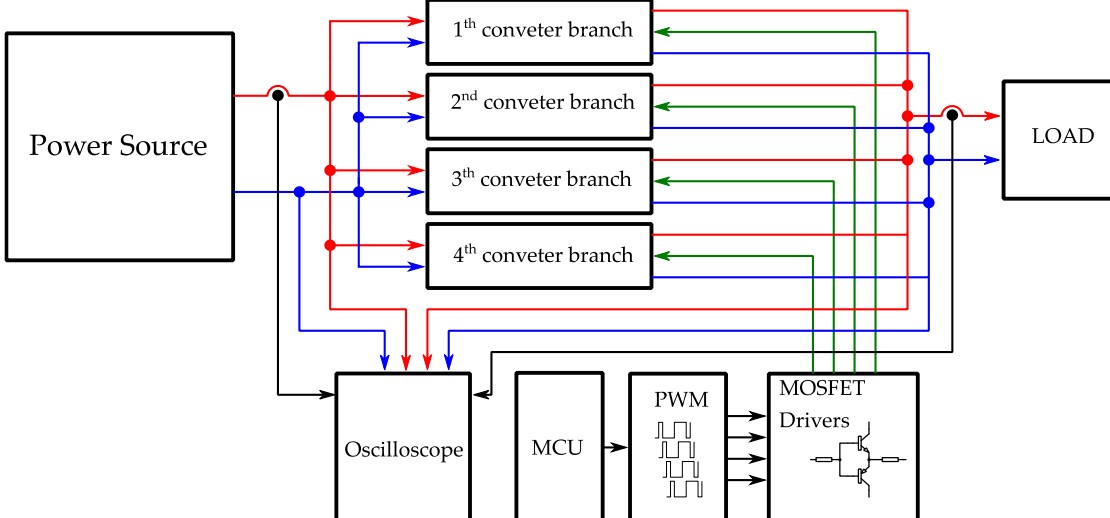

**Figure 10.** The block diagram depicting the experimental testing setup. Red arrows represent positive voltage, while blue arrows denote ground potential. Green arrows indicate the direction of control signals. Additionally, the blue and red arrows also depict the flow direction of electrical power.

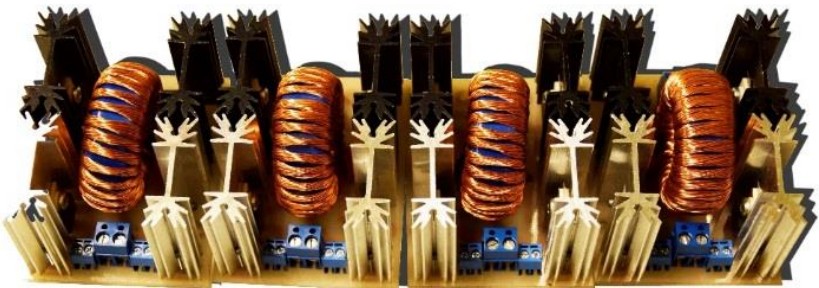

**Figure 11.** The constructed four-branch buck DC–DC pulse converter.

For generating control signals, an STM32F746ZG microcontroller was employed as the control unit. This 32-bit microcontroller, with a top CPU clock speed of 240 MHz, is aptly fast to handle all four branches of the converter. To safeguard the converter from any potential damage stemming from the microcontroller, a galvanic isolation approach was implemented. This galvanic separation is facilitated using optocouplers and isolated DC–DC converters on the power aspect, as depicted in Figure 12.

To ensure precise control, each converter branch's control signals are routed through distinct timers within the microcontroller. This separation is imperative as it allows for the essential phase shifting required to enable interleaved operation among the converter branches. An extra timer supervises the phase shift synchronization between the control signals. The timer's frequency can vary, depending on the count of converter branches and the control signals' switching frequency. It is noteworthy that this timer operates only when there is a change in the number of branches in the converter, ensuring efficient synchronization of control signals with the converter's configuration. As an illustration, Figure 13 depicts the voltage waveforms of the control signals originating from the employed MCU for the four branches of the converter.

In our experimental configuration, we employed the MOSFET IRF540N as the power switch in conjunction with the MBR20100CT power diode. Each converter branch featured a primary inductor with a value of 106 µH. The switching frequency for each of these converter branches was set to 200 kHz.

For a visual representation of this experimental setup, please refer to Figure 14. This diagram provides an overview of the physical arrangement and components used in our

testing environment. The coils within the converter are positioned on the underside of the designed printed circuit board, a strategic choice made to enhance accessibility for current and voltage probes. This placement facilitates a more straightforward and efficient connection to these probes, enabling precise measurement and monitoring of electrical parameters within the converter.

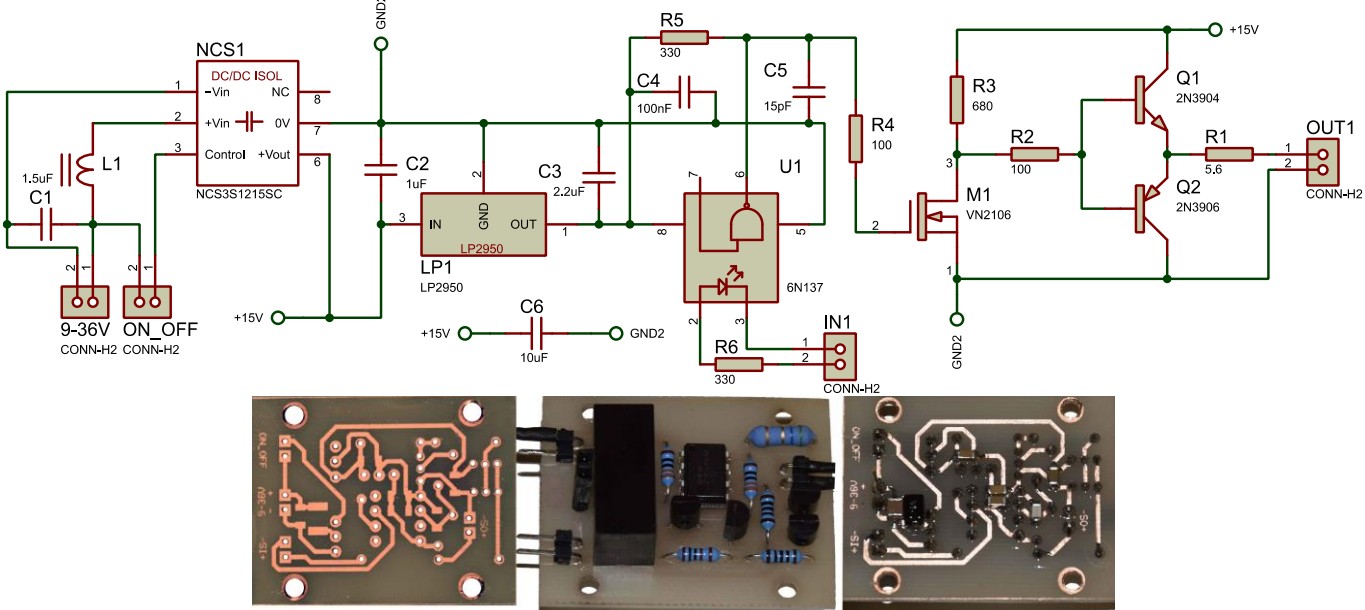

**Figure 12.** The proposed MOSFET driver, featuring galvanic isolation.

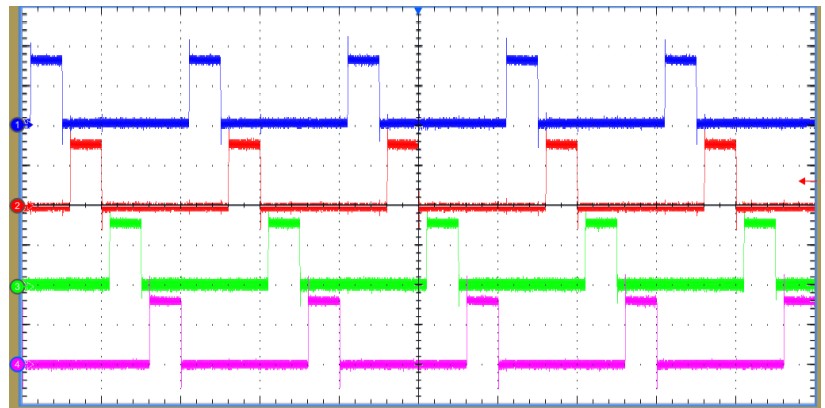

**Figure 13.** Control signals for MOSFETs drivers generated by microcontroller, 1: 2 V/div—1st branch, 2: 2 V/div—2nd branch, 3: 2 V/div—3rd branch, 4: 2 V/div—4th branch, time—2.5 µs/div.

The measurement procedure was executed as follows: Using a four-channel oscilloscope of the Tektronix DPO 7354 type, both input and output voltages, as well as currents, of the designed DC–DC converter were meticulously recorded.

To systematically vary the duty cycle, the microcontroller (MCU) implemented a stepwise adjustment with increments of 0.02 and a user-defined number of branches, all within a time interval of 100 ms.

Subsequently, the obtained data from the oscilloscope were imported into the MS Excel program for each branch configuration individually. These data sets underwent further processing and were presented in a unified graphical format, as exemplified in Figure 15. This graphical representation serves to illustrate the relationships and behavior of the converter under different branch configurations and duty cycle settings (input power).

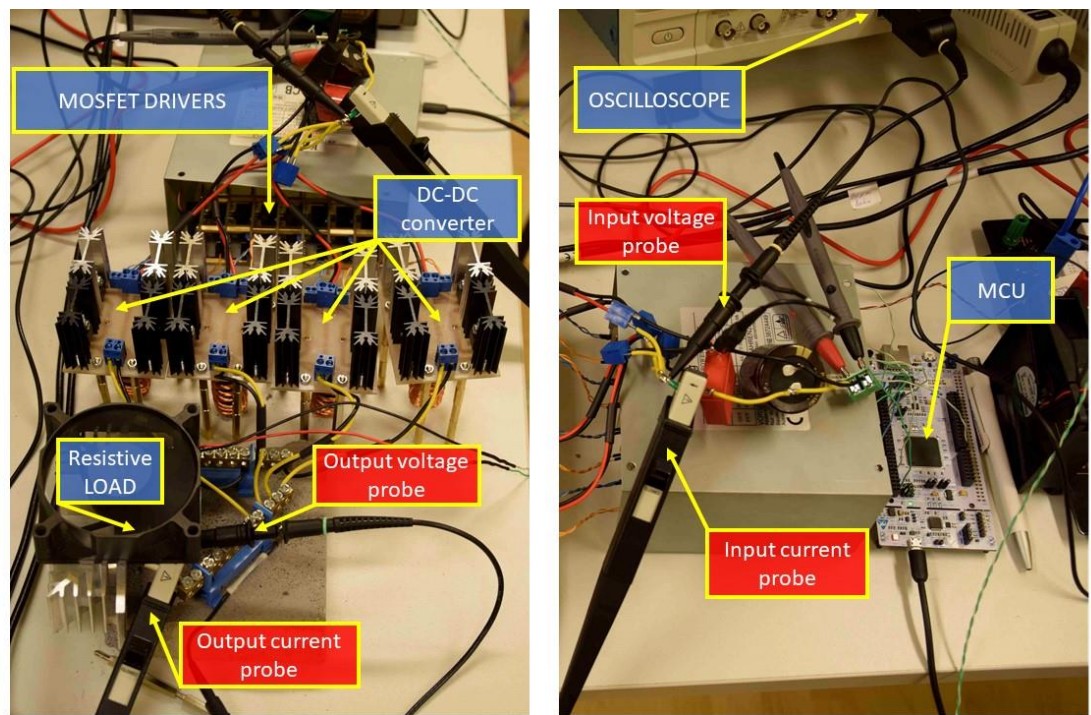

**Figure 14.** Experimental setup illustration.

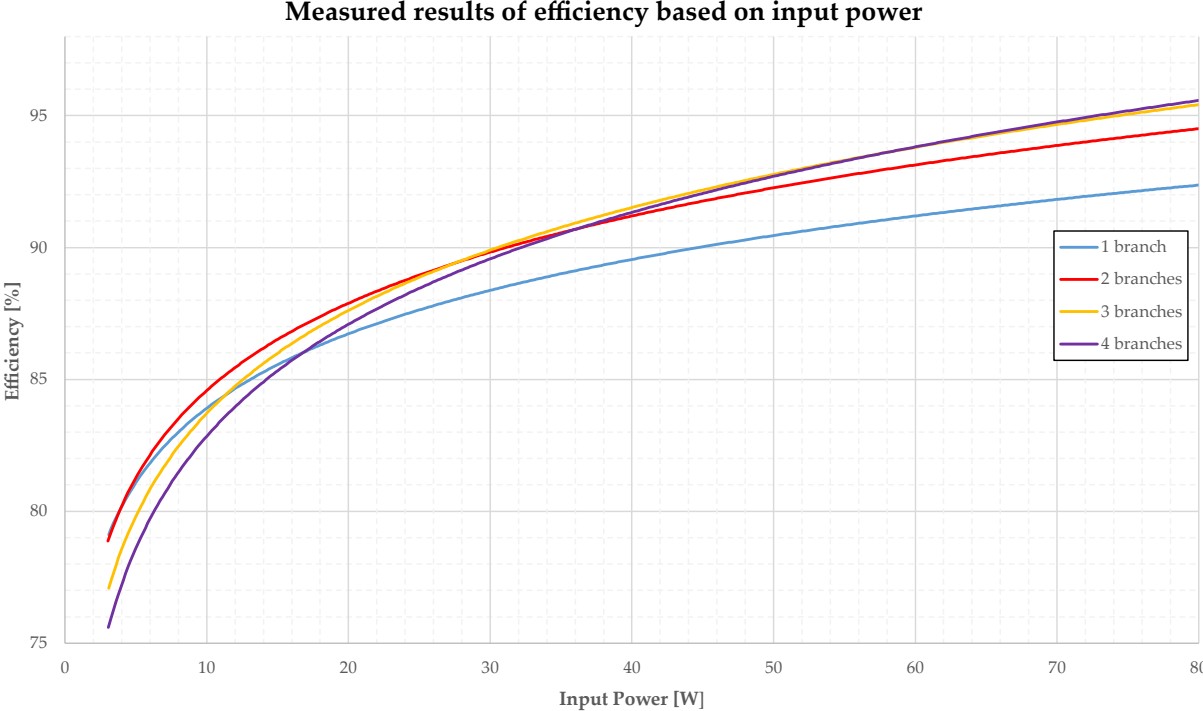

**Figure 15.** The efficiency curves obtained from experimental results, which showcase variations in efficiency with respect to different input power values and various numbers of converter branches.

These findings emphasize the important relationship between the number of branches and system performance, highlighting the importance of adaptability in the pursuit of optimal power delivery and efficiency across various operational scenarios.

This significant discovery reaffirms the notion that using a higher number of branches is not always the most suitable choice. Moreover, it corroborates the consistency between the simulation, calculation, and experimental results.

Based on these findings, it becomes possible to formulate an additional control algorithm, which can be integrated into the existing maximum power point tracking (MPPT) algorithms. Further details on this aspect are elaborated upon in the subsequent section.

These results hold practical implications for optimizing the performance of converters in various scenarios and can potentially lead to more efficient and adaptable control strategies.

In conclusion of this section, it is essential to highlight that throughout each phase (measurement, simulation, and calculation), the input voltage remained constant and was set at 30 V. A critical parameter in this context is the current, which significantly influences the optimal number of branches in the converter, as will be elaborated further. Thus, it is not solely the power value but rather the magnitude of the current that determines the calculated power and, consequently, the most suitable number of branches in the converter configuration. This distinction underscores the pivotal role of current in shaping the converter's operational efficiency and design considerations.

## 5. The Proposition of an Additional Algorithm That Dynamically Adjusts the Number of Branches Based on Efficiency Considerations

The fundamental cornerstone for achieving peak energy conversion efficiency, be it from a source like a photovoltaic panel, centers on the precise determination of its MPPT. Consequently, any control algorithm must align seamlessly with this fundamental requirement. Building upon the previously obtained results, it is feasible to enhance any existing algorithm by introducing efficiency monitoring and dynamically adjusting the number of converter branches in use.

To fulfill this imperative, a microcontroller, equipped with an A/D converter, undertakes the responsibility of measuring both input and output currents and voltages. These measured data serve as the foundational elements for crucial calculations, encompassing input power, output power, and the efficiency of the converter's power section. To ensure safe and accurate measurements, specialized modules were devised to galvanically isolate voltage and current measurements both before and after the converter. The design and realization of the input voltage and current measurement module are depicted in Figure 16, while the design and realization of the output voltage and current measurement module are illustrated in Figure 17.

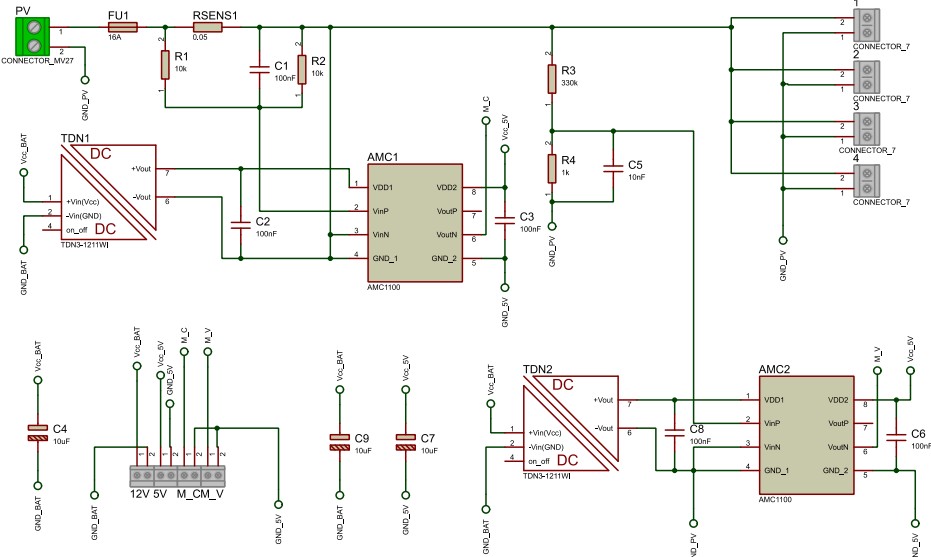

**Figure 16.** *Cont.*

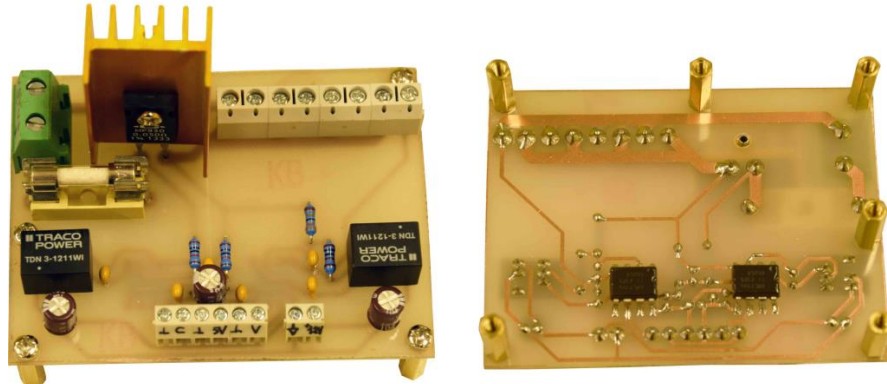

**Figure 16.** The schematic and implemented module for measuring input voltage and current with galvanic isolation.

**Figure 17.** The schematic and implemented module for measuring output voltage and current with galvanic isolation.

In light of the inherent losses within the converter, it becomes evident that it is not always optimal to activate all branches simultaneously. These insights underscore the paramount importance of developing a dynamic control strategy capable of adaptively modifying the number of active branches to maximize efficiency across varying operational conditions.

The initial segment of the proposed supplementary algorithm comprises the initialization and the initial startup of the converter with the maximum number of branches at MPP, as depicted in Figure 18. This phase is pivotal as it sets the stage for subsequent dynamic adjustments to the number of active branches based on real-time efficiency measurements and operating conditions.

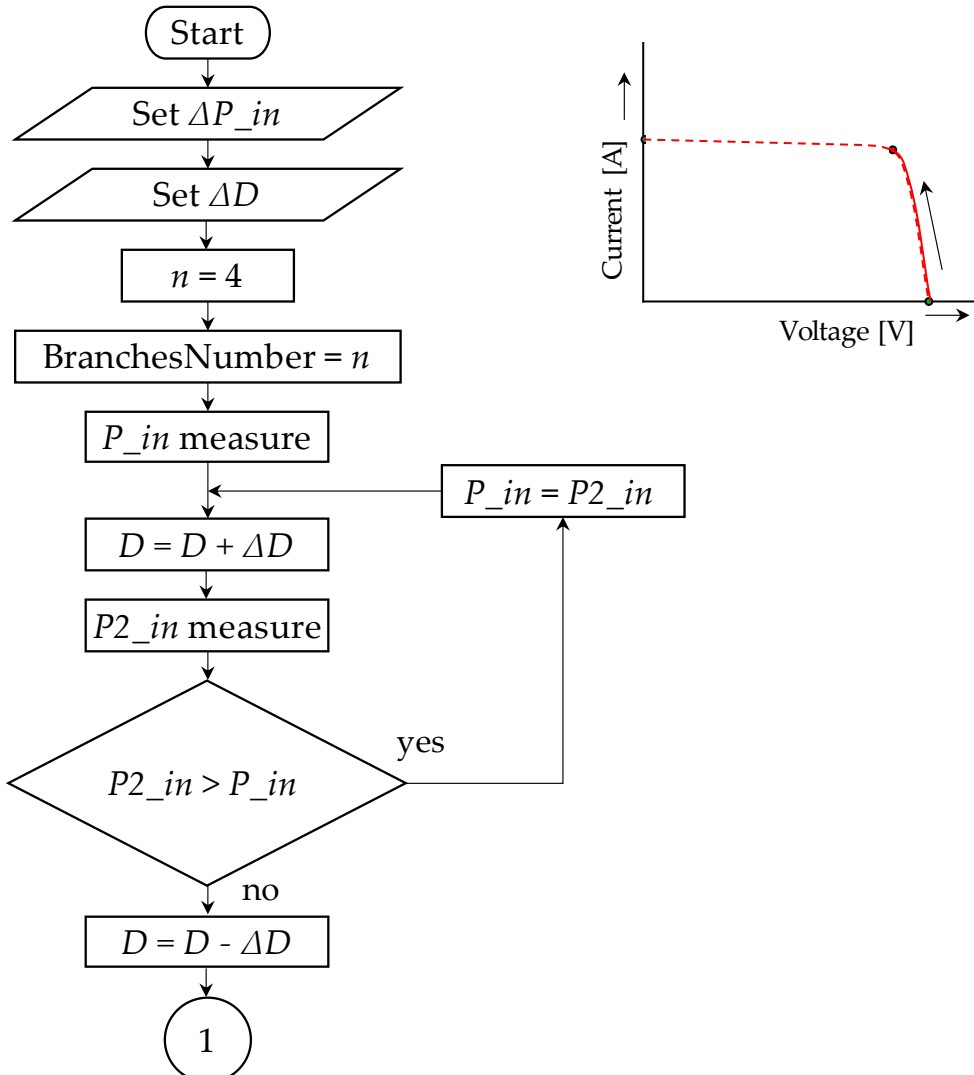

**Figure 18.** The developmental flowchart of the initial phase in the dynamic control strategy for the converter topology.

The following algorithm is a continuation of the preceding initialization phase. This developmental algorithm, as illustrated in Figure 19, operates in the following manner:

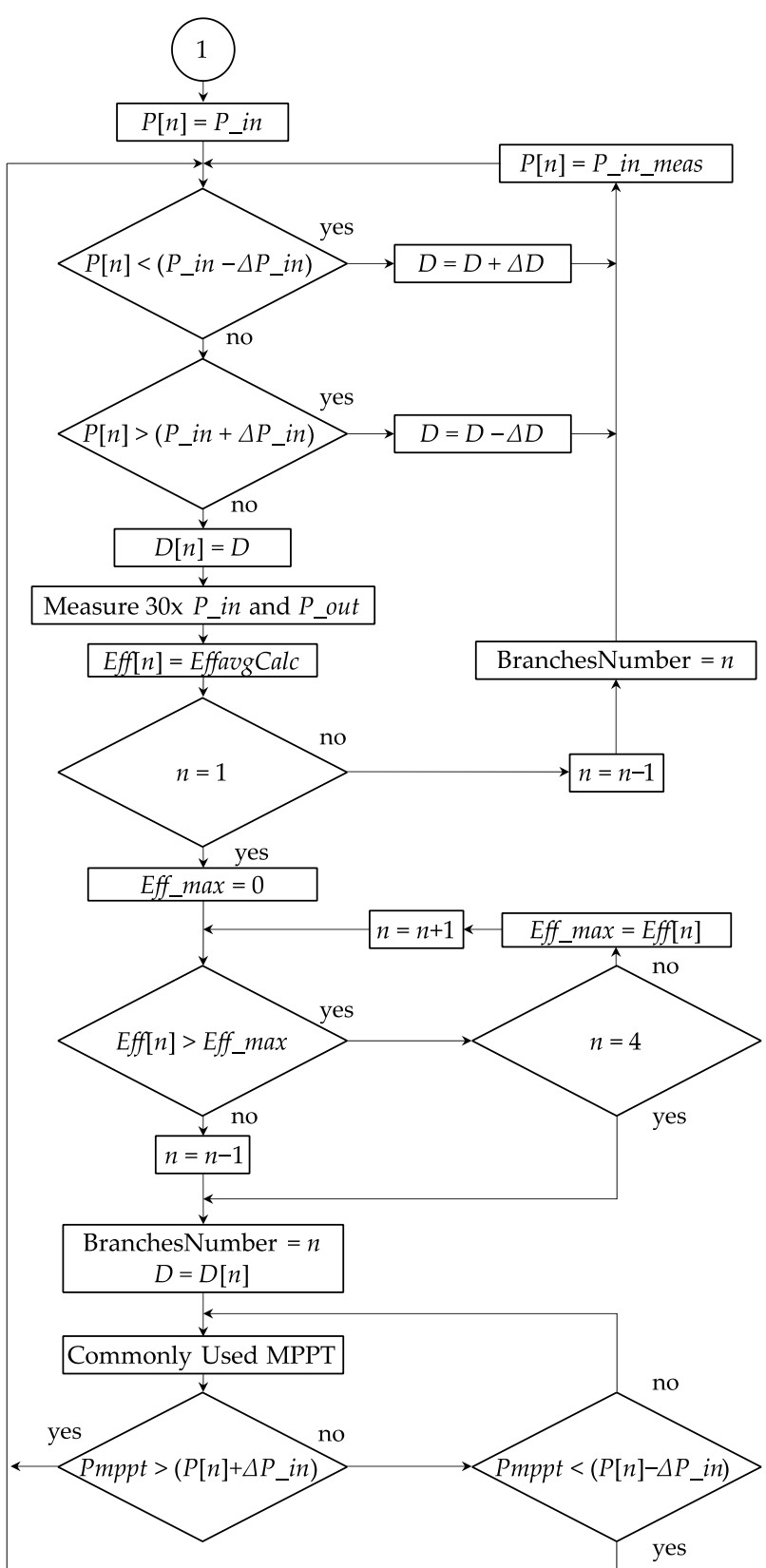

**Figure 19.** The developmental flowchart of the proposed algorithm for determining the required number of branches in the converter based on measured efficiency.

First, the current value of the input power is stored in a double-type variable array $P[n]$.

The variable $n$ is set to 4, as determined in the previous algorithm.

Next, it checks whether the power in the array exceeds the initial value plus or minus $\Delta P$. $\Delta P$ defines the hysteresis of the input power, specifying the range within which the measured input power can fluctuate without triggering a change in the number of branches or duty cycle adjustment.

After verifying the input power, the duty cycle value is recorded in the $D[n]$ array for the given number of branches.

Subsequently, input and output power measurements are conducted with averaging, and the efficiency for the current number of branches is calculated and stored in the $Eff[n]$ double-type variable.

Once the efficiency for the current number of branches is computed, it checks if the current number of branches is not equal to one. If not, it reduces the number of actively operating converter branches by one. The duty cycle is then adjusted to match the input power value as it was for a higher number of branches.

The process continues, systematically reducing the number of branches, recalculating efficiency, and storing the results.

After determining the efficiency for each number of branches, the algorithm selects the number of branches that achieved the highest measured efficiency.

Following this step, it proceeds to any standard MPPT algorithm. After achieving a new input power value that exceeds $\Delta P$, the proposed algorithm recommences the process of determining the most efficient number of branches.

This algorithm controls dynamic adjustments in the number of branches and duty cycle to maximize efficiency in real-time, adapting to changing input power conditions for improved energy conversion performance.

Another advantage of utilizing the proposed algorithm is that it does not necessitate prior knowledge of component parameters for determining the optimal number of branches through calculation. Instead, it actively measures the real-time efficiency, eliminating the need for predefined component values. This feature proves especially valuable in scenarios where component parameters might vary due to temperature fluctuations, potentially causing discrepancies between the calculated and actual required number of branches in the converter. In such cases, the algorithm can still provide a rough estimate of the required branch count, serving as a starting point for subsequent refinement through the algorithm's iterative process to precisely determine the optimal number of branches. This adaptability underscores the algorithm's versatility and effectiveness in optimizing converter configurations across varying conditions and component variations.

This type of algorithm was subsequently employed for practical validation to assess its real-world functionality. One of the notable advantages of this algorithm is that, in addition to incorporating previously designed MPPT algorithms, it provides the flexibility to dynamically adjust the number of branches, effectively altering the converter topology. This means that beyond enhancing efficiency through modern MPPT methods, it further elevates efficiency levels in scenarios where the input power to the converter spans a wide range. This improvement applies not only to high-power conditions but also extends its benefits to low-power operations. Consequently, it represents a valuable tool for optimizing converter performance across a broad spectrum of operating conditions.

*Results Obtained by Proposed Control Algorithm*

The results obtained through the application of the proposed algorithm capture the time intervals during which the number of branches in the converter decreases until it reaches one, followed by the MCU's decision to retain the branch with the highest average efficiency. These measurements were conducted across multiple input power levels, as illustrated in Figures 20–22.

Through the conducted measurements and application of the algorithm, a noteworthy finding emerged, emphasizing the significance of current as a crucial parameter in efficiency-based decision-making. This observation aligns with expectations, given that transient

losses are directly proportional to the transient resistance of a single branch and the square of the current flowing through it. This is substantiated by the measured variations in efficiency and the selection of the optimal number of branches, where it becomes evident that even when the input power is higher, the algorithm may evaluate a smaller number of branches (as seen in Figure 21) compared to cases with a higher number of branches (as in Figure 20) at lower power levels. These measurements encompassed diverse input voltage levels and maximum current values from the source, illustrating the algorithm's adaptive response to changing operational conditions while prioritizing efficiency optimization.

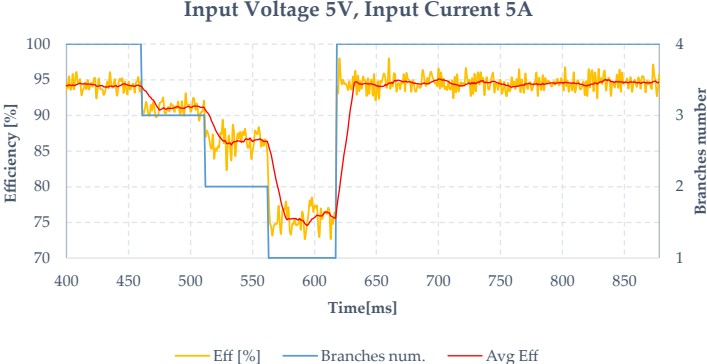

**Figure 20.** Results obtained from the use of the suggested algorithm indicate that a setup with four branches offers higher efficiency.

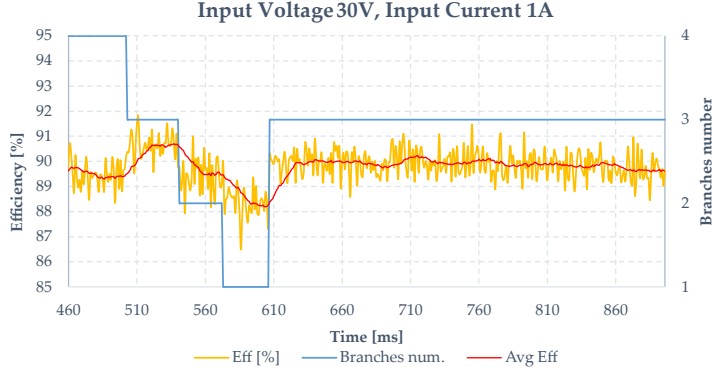

**Figure 21.** Measured results obtained through the utilization of the proposed algorithm, indicating that a configuration with three branches is more efficient.

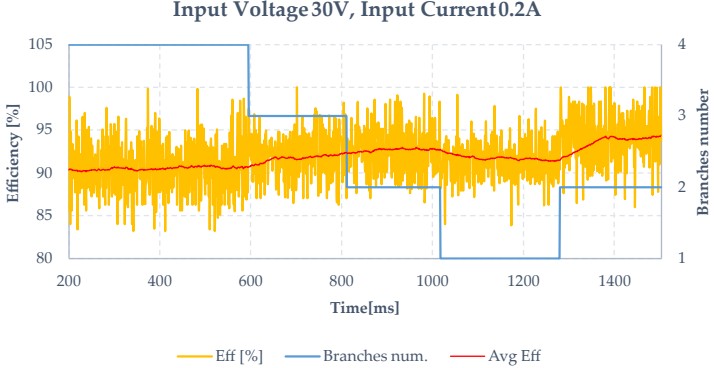

**Figure 22.** Measured results obtained through the utilization of the proposed algorithm, indicating that a configuration with two branches is more efficient.

The time required for the algorithm to make a decision depends on its capability to measure the same power level for each combination of branch numbers. As shown in

Figure 22, it is evident that the time needed to determine the more efficient number of branches is longer compared to Figures 21 and 22. This discrepancy arises because the proposed algorithm, in its initial phase after a change in the number of branches, endeavors to achieve the same power consumption as with the previous branch count. To expedite this process, faster A/D converters and higher-performance MCUs can be employed, which would reduce the decision-making time, enhancing the algorithm's responsiveness in dynamic situations.

These visual representations effectively depict the algorithm's dynamic decision-making process in response to varying input power conditions, showcasing its ability to adapt and optimize the converter configuration for enhanced efficiency across different operational scenarios.

## 6. Conclusions

In summary, our investigation reveals the critical importance of adaptability in maximizing the efficiency of multi-branch buck DC–DC converters used in photovoltaic systems. The inherent losses within the converter underscore that activating all branches simultaneously is not always optimal. This necessitates the development of dynamic control strategies capable of adjusting the number of active branches to suit diverse operational conditions. Furthermore, our findings emphasize that current, rather than power value, plays a pivotal role in shaping converter efficiency, highlighting the significance of precise current control in design and operation.

Our algorithm, demonstrated through visual representations, showcases its ability to dynamically optimize the converter configuration under varying input power conditions, offering enhanced efficiency across different operational scenarios. These insights suggest that employing a higher number of branches is not always ideal, aligning with consistent results across simulation, calculation, and experimentation. The proposed control algorithm, designed to seamlessly integrate with existing MPPT strategies, holds practical implications for optimizing converter performance in various scenarios.

Looking forward, addressing the overarching challenge of photovoltaic panel efficiency remains paramount. Future research should focus on refining both direct and indirect MPPT methods, potentially harnessing advanced technologies like AI. The innovative algorithm developed for DC–DC converters also warrants further adaptability testing in real-world scenarios. Ultimately, the integration of these findings into broader sustainable energy systems and renewable sources promises a more efficient and sustainable energy future.

## 7. Discussion

The inherent losses within the converter reveal that activating all branches simultaneously is not always optimal. This underscores the need for a dynamic control strategy to adjust the number of active branches adaptively, maximizing efficiency across various operational conditions. The importance of this adaptability becomes evident when optimizing power delivery and efficiency in diverse scenarios.

Throughout the measurement, simulation, and calculation phases, the input voltage remained constant. However, the critical parameter is the current, significantly influencing the optimal number of branches. Current, rather than power value, determines the calculated power and, consequently, the suitable branch count, emphasizing its pivotal role in shaping the converter's efficiency and design.

The visual representations demonstrate the algorithm's dynamic decision-making process, adapting to varying input power conditions and optimizing the converter configuration for enhanced efficiency in diverse scenarios. These findings underscore the relationship between the number of branches and system performance, emphasizing adaptability for optimal power delivery and efficiency across different operational circumstances.

This discovery reinforces that employing a higher number of branches is not always ideal and aligns with the consistency between simulation, calculation, and experimental

results. These insights enable the development of an additional control algorithm, seamlessly integrating with existing MPPT strategies. These results offer practical implications for optimizing converter performance in various scenarios, potentially leading to more efficient and adaptable control strategies.

The efficiency of photovoltaic panels remains a central challenge, emphasizing the need for enhanced materials and design research. With the paper's focus on maximum power point tracking (MPPT), future studies could refine both direct and indirect tracking methods, leveraging advanced technologies like AI. Additionally, there's potential in further developing the innovative algorithm for DC–DC converters, ensuring its adaptability in diverse real-world scenarios. Ultimately, integrating these findings with broader sustainable energy systems and other renewable sources can pave the way for a more efficient and sustainable energy future.

**Author Contributions:** Conceptualization, M.P.; methodology, T.V.; software, M.B.; validation, J.M. and D.K.; formal analysis, I.K. and M.B.; investigation, M.P. and M.B.; resources, M.B. and D.K.; data curation, M.B.; writing—M.B. and M.P.; writing—M.B., M.P. and T.V.; visualization, M.B. and M.P.; supervision, M.B.; project administration, M.B. and M.P.; funding acquisition, M.P. All authors have read and agreed to the published version of the manuscript.

**Funding:** This work was supported by the Slovak Research and Development Agency under contract No. APVV-19-0576 and APVV-21-0312.

**Institutional Review Board Statement:** Not applicable.

**Informed Consent Statement:** Not applicable.

**Data Availability Statement:** Data in this paper are publicly unavailable due to privacy restrictions.

**Acknowledgments:** All support given during the work on this paper is covered by the author's contribution or funding sections.

**Conflicts of Interest:** The authors declare no conflict of interest.

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
