# Peer review of "Efficiency Optimization in Multi-Branch Converters through Dynamic Control"

_sustainability, doi:10.3390/su152216032_

Round 1

Reviewer 1 Report

Comments and Suggestions for Authors

The article presents a multi-branch buck converter designed for solar panels with automatic maximum power point tracking (MPPT). The article is logically structured and provides a thorough description of the theory, although there could be more consistent references to the literature used in various sections.

The authors derive specific mathematical relationships for calculating instantaneous currents and voltages in the circuit, which are then used to create a simulation model and theoretically calculate the efficiency curve of the converter for various numbers of branches at a given power level. The article subsequently discusses the implementation of the proposed converter and compares the measured values with the simulation results, revealing no significant differences.

To improve the quality of the article, the following shortcomings need to be corrected:

- The introduction repeatedly mentions "invertor," but the rest of the manuscript deals with "convertor." This inconsistency affects the overall quality of the article.

- Equation (2) needs to be double-checked. It refers to source [6], but no such formula was found in that source. Is solar radiation intensity G in volts? The formula seems to indicate so. Additionally, in the text, "Umpp" is mentioned, but the formula contains "Vmpp." The text also contains an uppercase "U," while the formula and the sentence before use a lowercase "u." It's unclear if these parameters are identical or not. If they are not, clarification is needed regarding what "U" represents when it is not in the formula.

- In Figure 4, current I1 and I2 need more explicit explanations. Are they minimal and maximal currents, or do they represent the current over branch 1 and branch 2? The text provides some hints, but these current designations need explicit clarification.

- Figure 17 is atypically divided by the page. The schematic and implemented module should be separated into two distinct figures. Similarly, Figure 18 should follow the same format.

- Where exactly are the resource 7 – 14? They are not referenced anywhere in the text.

Author Response

Response to Reviewer 1 Comments

Dear Reviewer,

Thank you very much for your time reviewing our manuscript on Efficiency Optimization in Multi-Branch Converters through Dynamic Control.

We truly appreciate your valuable comments which in turn modifies and contributes additional knowledge to our proposed paper to make it clearer and more valuable solution.

Regarding your specific comments, I would like to respond as follows:

1) The introduction repeatedly mentions "invertor," but the rest of the manuscript deals with "convertor." This inconsistency affects the overall quality of the article..

Thank you very much for your insightful feedback on my manuscript. I appreciate your attention to detail in pointing out the inconsistency between the terms "invertor" and "convertor." Upon reviewing the sections you mentioned, I agree that aligning these terms will enhance the clarity and coherence of the article. I will make the necessary revisions to address this issue and ensure consistency throughout the manuscript.

2) Equation (2) needs to be double-checked. It refers to source [6], but no such formula was found in that source. Is solar radiation intensity G in volts? The formula seems to indicate so. Additionally, in the text, "Umpp" is mentioned, but the formula contains "Vmpp." The text also contains an uppercase "U," while the formula and the sentence before use a lowercase "u." It's unclear if these parameters are identical or not. If they are not, clarification is needed regarding what "U" represents when it is not in the formula

The necessary revisions have been made to address your concerns: Equation (2) has been corrected to reflect the appropriate source, the unit for solar radiation intensity 'G' has been clarified, and the inconsistency between "Umpp" and "Vmpp" has been resolved for uniformity. Additionally, the usage of uppercase "U" and lowercase "u" has been standardized throughout the document with added clarification on their specific meanings.

3) In Figure 4, current I1 and I2 need more explicit explanations. Are they minimal and maximal currents, or do they represent the current over branch 1 and branch 2? The text provides some hints, but these current designations need explicit clarification.

In response to your review note regarding Figure 4, We have added explicit explanations to the text. This added clarification directly addresses the ambiguity previously noted in the figure's interpretation.

4) Figure 17 is atypically divided by the page. The schematic and implemented module should be separated into two distinct figures. Similarly, Figure 18 should follow the same format.

In response to your note about Figure 17 and Figure 18, We agree with your suggestion for improvement. After the completion of the revision tracking process, We will reorganize these figures.

5) Where exactly are the resource 7 – 14? They are not referenced anywhere in the text.

Citations have been added to the text in the Introduction chapter, where we describe various methods for searching the MPP point that have already been studied and published.

Reviewer 2 Report

Comments and Suggestions for Authors

The article presents the dynamic control-based efficiency optimization method. It is an interesting article. I have the following suggestions for improvement.

1. The abstract should highlight the brief contributions and expectations from this article, and the method to achieve them, instead of highlighting previous methods, etc.

2. Some information such as Fig. 1 can be considered good for a thesis but not for an article. In articles, it is better to make the discussion brief and concise.

3. The overview of the multistage DC-DC converters and relevant mathematical modeling for various operation modes should be included. The following articles can be relevant in this regard.

https://doi.org/10.1155/2020/5892190

https://doi.org/10.5755/j01.eie.26.3.25763

4. The quality of images and graphs should be improved. 

5. It is better to include the relevant equation numbers in the flowcharts for better understanding.

6. The discussion section should be followed/replaced by the conclusion containing the significant results and achievements of the work.

7. The novel contributions should be highlighted in the introduction section explicitly. 

Thanks

Author Response

Response to Reviewer 2 Comments

Dear Reviewer,

Thank you very much for your time reviewing our manuscript on Efficiency Optimization in Multi-Branch Converters through Dynamic Control.

We truly appreciate your valuable comments which in turn modifies and contributes additional knowledge to our proposed paper to make it clearer and more valuable solution.

Regarding your specific comments, I would like to respond as follows:

1) The abstract should highlight the brief contributions and expectations from this article, and the method to achieve them, instead of highlighting previous methods, etc.

We appreciate your feedback regarding the abstract of our article. We have carefully considered your suggestion, and we have made the necessary revisions to align with your recommendation. The updated abstract now focuses on succinctly highlighting the key contributions and expectations from this article, along with outlining the methodology employed to achieve them. We have ensured that previous methods and related details have been appropriately condensed or omitted to provide a more concise and impactful summary. We believe these changes enhance the clarity and effectiveness of our abstract, and we thank you for your valuable input.

2) Some information such as Fig. 1 can be considered good for a thesis but not for an article. In articles, it is better to make the discussion brief and concise.

Based on your recommendation, Figure 1 has been removed from the contribution. The remaining figures have also been renumbered in the text.

3) The overview of the multistage DC-DC converters and relevant mathematical modeling for various operation modes should be included. The following articles can be relevant in this regard. https://doi.org/10.1155/2020/5892190 https://doi.org/10.5755/j01.eie.26.3.25763

Thank you for your valuable suggestion regarding the inclusion of an overview of multistage DC-DC converters and their mathematical modeling. Following your recommendation, we have added the suggested articles to the references section.

4) The quality of images and graphs should be improved.

Thank you for your notice. Most of the figures are in vector format, except for pictures of device parts and the laboratory setup. Therefore, we are unsure to which figures you are referring.

5) It is better to include the relevant equation numbers in the flowcharts for better understanding.

While considering the inclusion of equation numbers in flowcharts, it's worth noting that doing so may not necessarily enhance understanding. In some cases, equation numbers in flowcharts can lead to visual clutter and may obscure the flow and clarity of the diagram. Flowcharts are primarily designed to provide a clear visual representation of processes, steps, or relationships, and adding equation numbers can potentially distract from their primary purpose. Instead, it might be more effective to keep equations and their corresponding numbers separate, possibly in a different section or referenced in the caption or accompanying text. This way, readers can still access the equations while maintaining the flowchart's visual simplicity and comprehensibility.

6) The discussion section should be followed/replaced by the conclusion containing the significant results and achievements of the work.

Thank you for bringing up your concern about the structure of our paper. We have reviewed your suggestion, and we would like to inform you that we have made the necessary changes to the paper. We have ensured that the discussion section is now followed by the conclusion, as per your recommendation. In the conclusion section, we have included a concise summary of the significant results and achievements of our work. We believe this adjustment enhances the overall flow and clarity of our paper, and we appreciate your valuable feedback..

7) The novel contributions should be highlighted in the introduction section explicitly.

Thank you for your feedback. We have carefully considered your suggestion, and we are pleased to inform you that the requested changes have been made to the introduction section. We have explicitly highlighted the novel contributions of our research in accordance with your recommendation. We believe these enhancements will significantly improve the clarity and impact of our paper, and we appreciate your valuable input.

Reviewer 3 Report

Comments and Suggestions for Authors

The paper focuses on enhancing the efficiency of non-isolated multi-branch buck DC-DC converters, critical components in photovoltaic power systems. The central challenge addressed is achieving Maximum Power Point Tracking (MPPT) to optimize energy conversion by dynamically adjusting the number of active branches. The study investigates various methods for MPPT and introduces an innovative algorithm that dynamically adapts the active branches based on efficiency considerations. It emphasizes the role of current in shaping converter efficiency and proposes an adaptive control strategy. The results reveal that activating all branches simultaneously isn't always optimal, highlighting the need for dynamic control to maximize efficiency under varying operational conditions. The study showcases a strong relationship between the number of branches and system performance, emphasizing adaptability for optimal power delivery and efficiency across diverse circumstances. The findings underscore the significance of current in influencing optimal branch count and, by extension, converter efficiency. Additionally, the paper offers practical implications for optimizing converter performance, potentially leading to more efficient and adaptable control strategies.

My Comments: 

1)  Could the paper provide a more detailed explanation of the dynamic control algorithm introduced for adjusting the number of active branches? 

2) While the paper mentions efficiency gains resulting from the dynamic control algorithm, could it provide quantitative data or a comparison with static control methods to quantify the improvements achieved?

3)  Given that the study emphasizes varying operational conditions, were there any variations or uncertainties encountered during measurements or simulations, and how were these accounted for in the results and analysis?

4) Did the study consider the impact of non-ideal components (e.g., losses, parasitics) in the converter, and how do these affect the effectiveness of the dynamic control algorithm?

5) Were any challenges encountered or limitations faced during the practical validation of the mathematical and simulation models, and how were these addressed or mitigated?

6)  How do the algorithm and the efficiency of the converter respond to fluctuations in the input voltage, and was this aspect thoroughly considered in the simulations and practical validation?

7) How feasible is it to implement the proposed adaptive control strategy in real-time systems, and were any real-time considerations addressed in the study?

8) Does the study touch upon the economic feasibility or cost implications associated with implementing the proposed adaptive control strategy, especially in large-scale photovoltaic systems?

9)  Could the paper suggest potential future research directions or improvements to the algorithm or methodology proposed, considering advancements in technology and emerging challenges in photovoltaic systems?

Author Response

Response to Reviewer 3 Comments

Dear Reviewer,

Thank you very much for your time reviewing our manuscript on Efficiency Optimization in Multi-Branch Converters through Dynamic Control.

We truly appreciate your valuable comments which in turn modifies and contributes additional knowledge to our proposed paper to make it clearer and more valuable solution.

Regarding your specific comments, I would like to respond as follows:

1)  Could the paper provide a more detailed explanation of the dynamic control algorithm introduced for adjusting the number of active branches? 

Detailed algorithm for adjusting the number of active branches is already presented in Figure 19. as initial phase and Figure 20. presents dynamic control based on measured efficiency and also in text description below Figure 20. In a more detailed manner, providing the source code of the microcontroller program might be possible; however, it could compromise the clarity of the algorithm itself. Of course, the source code itself is accessible upon request.

2) While the paper mentions efficiency gains resulting from the dynamic control algorithm, could it provide quantitative data or a comparison with static control methods to quantify the improvements achieved?

Such a comparison data are presented in Figure 6, Figure 10 and Figure 16. According to these figures, efficiency gains resulting from the dynamic control algorithm depends on the differences on Input power. Of course, or measured and simulated data are provided on request.

3)  Given that the study emphasizes varying operational conditions, were there any variations or uncertainties encountered during measurements or simulations, and how were these accounted for in the results and analysis?

Within the differences between simulation and measurements, only negligible variances were observed, which can be noticed, for example, by comparing images Figure 6. and Figure 16. As these differences were minimal, there was no need to consider them in the results or analysis.

4) Did the study consider the impact of non-ideal components (e.g., losses, parasitics) in the converter, and how do these affect the effectiveness of the dynamic control algorithm?

Because the algorithm is based on measured values, it is not influenced by component inaccuracies or parasitic elements of the converter itself. The goal of the converter is not to calculate precise efficiency but to determine whether changes in the duty cycle or the number of active branches have led to an increase or decrease in efficiency.

5) Were any challenges encountered or limitations faced during the practical validation of the mathematical and simulation models, and how were these addressed or mitigated?

During the validation of mathematical and simulation models, no significant challenges or limitations were identified. Of course, the simulation model did not work perfectly right from the start, but most of the shortcomings were attributed to human error and were rectified without major complications.

6)  How do the algorithm and the efficiency of the converter respond to fluctuations in the input voltage, and was this aspect thoroughly considered in the simulations and practical validation?

Because the algorithm is designed for iterative stepwise changes, fluctuations in input voltage did not cause system instability. On the other hand, a drawback of this algorithm design can be considered the slower determination of optimal parameters compared to other methods. An alternative algorithm might allow parameter changes in a more drastic, stepwise manner, but this would come at the expense of system stability. This is precisely why the gradual iterative change algorithm appears to be more advantageous.

7) How feasible is it to implement the proposed adaptive control strategy in real-time systems, and were any real-time considerations addressed in the study?

The proposed converter is primarily intended for application in photovoltaic panels. The conditions and parameters of photovoltaic panels typically do not change in milliseconds, as is the case in real-time systems. Therefore, this study did not address the control strategy for real-time systems. Theoretically, it is possible to speed up the iteration cycle in the controller's program to make the algorithm usable in real-time systems, but this issue is beyond the scope of the article itself.

8) Does the study touch upon the economic feasibility or cost implications associated with implementing the proposed adaptive control strategy, especially in large-scale photovoltaic systems?

Since the output of a photovoltaic panel is significantly influenced by the intensity of sunlight, a converter is essential for practical energy usage from the panel. The price of converters can vary significantly, and one of the parameters to consider is their efficiency. The proposed converter, as it appears, seems to be more cost-effective than existing converters with comparable efficiency. However, this, of course, depends on the specific components used. It's worth noting that a comprehensive economic feasibility study wasn't conducted in the scope of this article. The article primarily presents a result of a proof of concept for the converter itself, so more in-depth analysis of economic feasibility falls beyond the scope of this article.

9)  Could the paper suggest potential future research directions or improvements to the algorithm or methodology proposed, considering advancements in technology and emerging challenges in photovoltaic systems?

Future research directions and considering advancements in technology were added in discussion.

Round 2

Reviewer 2 Report

Comments and Suggestions for Authors

OK for publication

Reviewer 3 Report

Comments and Suggestions for Authors

Accept in present form.